# Comparative Genomic Analysis of Antimicrobial-Resistant *Escherichia coli* from South American Camelids in Central Germany

**DOI:** 10.3390/microorganisms10091697

**Published:** 2022-08-24

**Authors:** Belén González-Santamarina, Michael Weber, Christian Menge, Christian Berens

**Affiliations:** Friedrich-Loeffler-Institut, Institute of Molecular Pathogenesis, Naumburger Str. 96a, 07743 Jena, Germany

**Keywords:** South American camelids, antimicrobial resistance, virulence factor, pathovar, genome analysis, *E. coli*, Germany

## Abstract

South American camelids (SAC) are increasingly kept in Europe in close contact with humans and other livestock species and can potentially contribute to transmission chains of epizootic, zoonotic and antimicrobial-resistant (AMR) agents from and to livestock and humans. Consequently, SAC were included as livestock species in the new European Animal Health Law. However, the knowledge on bacteria exhibiting AMR in SAC is too scarce to draft appropriate monitoring and preventive programs. During a survey of SAC holdings in central Germany, 39 *Escherichia coli* strains were isolated from composite fecal samples by selecting for cephalosporin or fluoroquinolone resistance and were here subjected to whole-genome sequencing. The data were bioinformatically analyzed for strain phylogeny, detection of pathovars, AMR genes and plasmids. Most (33/39) strains belonged to phylogroups A and B1. Still, the isolates were highly diverse, as evidenced by 28 multi-locus sequence types. More than half of the isolates (23/39) were genotypically classified as multidrug resistant. Genes mediating resistance to trimethoprim/sulfonamides (22/39), aminoglycosides (20/39) and tetracyclines (18/39) were frequent. The most common extended-spectrum-β-lactamase gene was *bla*_CTX-M-1_ (16/39). One strain was classified as enteropathogenic *E. coli*. The positive results indicate the need to include AMR bacteria in yet-to-be-established animal disease surveillance protocols for SAC.

## 1. Introduction

Alpacas (*Vicugna pacos*) and llamas (*Lama glama*) [1], camelids of the suborder *Tylopoda*, are grouped under the designation South American camelids (SAC). They differ taxonomically, physiologically and behaviorally from ruminants (suborder *Ruminantia*) [2]. SAC have become very popular in Europe, including Germany, as evidenced by the steadily increasing number of these animals [3,4]. SAC are not only kept for wool production or for breeding [1,3,4]; other common occupations include landscape management or serving as animals for assisted therapy, trekking tours, exhibition or as pets [3,4]. This puts them frequently in close contact with humans. Although the Commission Implementing Regulation (EU) 2018/1882 recognizes SAC as carriers of different zoonotic agents with the potential to transmit them to other livestock species and to humans [5], the health and microbial status of these animals is not yet systematically recorded in Germany [3]. The resulting knowledge gap includes surveillance data on antimicrobial-resistant microorganisms, which, according to the recently enacted European Animal Health Law (EU Regulation 2016/429), should be treated as a transmissible disease.

Antimicrobial resistance (AMR) in bacteria is considered one of the greatest threats to human and animal health [6]. For 2019, predictive statistical models estimated 4.95 million human deaths worldwide to be associated with bacterial AMR, including 1.27 million deaths attributable to this phenomenon [7]. AMR transmission pathways are complex and diverse; the potential routes are from livestock to farmers and consumers, from pets to their owners and vice versa [8]. *Escherichia coli* is a standard indicator species used in human and veterinary public health to monitor AMR in Gram-negative bacteria of the gut microbiota [9,10,11]. For SAC, only a few studies in South America [12,13] and the UK [14] have analyzed the AMR phenotype of *E. coli* strains isolated from clinical cases thus far. In Germany, knowledge of the prevalence of AMR in SAC is also sparse. A recent pilot study conducted in central Germany, which selected for *E. coli* resistant to fluoroquinolones or cephalosporins, produced prevalence data for phenotypic resistance to cephalosporins among AMR *E. coli* from SAC that were similar to values obtained for other livestock species [4]. 

The genome sequences of AMR *E. coli* isolates can provide valuable information on virulence factors, the presence of pathovars, such as enteroaggregative (EAEC), enteroinvasive (EIEC), enteropathogenic (EPEC) or Shiga toxin-encoding (STEC) strains [15]. It is also imperative to identify the AMR determinants that are transferable to other enteric bacteria [9] and the mobile genetic elements mediating these transfer events. Such data can inform on strain epidemiology, cluster formation and overlap with strains isolated from human clinical cases or diseased animals, thus contributing to risk assessment for animal and human health. No previous study on AMR in SAC had employed whole-genome sequencing (WGS) for typing and characterizing *E. coli* strains and for the detection of antimicrobial resistance genes and mutations. Two *E. coli* strains (one STEC, one atypical EPEC), isolated from alpaca feces in Peru, had been whole-genome sequenced without further characterization [16]. The genomes of 38 strains isolated from SAC in Bolivia, Peru and the USA [17,18] have been deposited in Enterobase (accessed on 16 February 2022; [19]). 

To gain first insights into the population structure of AMR *E. coli* from SAC in Germany, 39 *E. coli* strains, isolated in 2019 during a study in central Germany [4], were subjected to short-read WGS. The resulting sequences were used to characterize the strains with respect to their phylogeny, pathogenic potential and AMR determinants. They were then compared to those of strains from other animals and humans deposited in sequence databases. Assigning the German SAC strains within the broader context of the *E. coli* population to clonal lineages and to pathovars associated with high-risk clones at the international level informs future surveillance studies but also leverages the development of targeted monitoring and preventive medicine measures.

## 2. Materials and Methods

### 2.1. Strain Selection

For this study, we selected 39 *E. coli* strains from a collection of 63 isolates [4], which were tested for susceptibility to twelve antibiotics and phylogenetically typed using a MLVA approach [20]. The strains were isolated from composite fecal samples of fresh droppings from alpaca and llama groups from 24 holdings located in the three German federal states of Saxony, Saxony-Anhalt and Thuringia. To ensure maximum diversity of the *E. coli* strain collection selected for sequencing, isolates were chosen on the basis of their respective (i) holding, (ii) MLVA profile and (iii) antimicrobial resistance profile. We selected at least one strain per holding and a broad spectrum of diverse MLVA and antimicrobial resistance profiles.

### 2.2. Whole-Genome Sequencing

The strains were grown overnight at 37 °C in 5 mL Luria–Bertani broth (Carl Roth GmbH, Karlsruhe, Germany). Genomic DNA was purified using the DNeasy^®^ UltraClean^®^ Microbial Kit (Qiagen GmbH, Hilden, Germany). DNA concentrations were determined using a NanoDrop^TM^ One/OneC Microvolume UV-Vis Spectrophotometer (Thermofisher, Dreieich, Germany). DNA sequencing libraries were prepared, and paired-end sequencing was performed by Eurofins Genomics Europe Sequencing (Illumina NovaSeq, 2 × 150 bp; Constance, Germany) or LGC Genomics (Illumina NovaSeq, 2 × 250 bp; Berlin, Germany). The sequencing facility/platform had no impact on sequencing quality. The genome assemblies of reads from both companies yielded high-quality contigs (LGC average N50 = 169 kb, Eurofins average N50 = 148 kb).

We performed a bioinformatic analysis of the strains using the “in-house” WGSBAC pipeline (https://gitlab.com/FLI_Bioinfo/WGSBAC, (accessed on 1 February 2022)). Illumina raw reads were subjected to quality control using FastQC (v. 0.11.7) (and the coverage was calculated using an adapted script (https://github.com/raymondkiu/fastq-info/blob/master/fastq_info_3.sh, (accessed on 1 June 2021)). The reads were de novo assembled using SPAdes (v. 3.15) [21] and evaluated with QUAST (v. 5.0.2) [22] with standard settings. Gene annotation was performed with Prokka (v. 1.14.5) [23]. The pipeline used Kraken 2 (v. 1.1) to identify contaminations and the Kraken2DB database to classify both reads and contigs [24]. The genes and chromosomal mutations encoding resistance were detected using AMRFinderPlus software (v. 3.10) [25]. Furthermore, Abricate (v. 1.0.1) was used with ResFinder (v. 3.2) [26], CARD (v. 3.0.8) [27] and the NCBI databases for resistance gene detection. Gene maps were generated for selected gene regions with the gggenes R package (v. 0.4.1). Abricate was also used in conjunction with the Virulence Factor Database (VFDB) [28] and Virulence Finder (v. 2.0) [29] for the prediction of virulence-associated genes. For the identification of extraintestinal pathogenic *E. coli* (ExPEC), all isolates were screened for the presence of five virulence markers: *papA* or *papC*, *sfa*/*foc*, *afa*/*dra*, *iutA* and *kps*MT II. Isolates positive for ≥2 markers were classified as ExPEC [30]. For the identification of potential uropathogenic *E. coli* (UPEC), all isolates were screened for the presence of the genes *chuA*, *fyuA*, *vat* or *yfcV* [31]. Strains positive for three or more of the four genes were considered UPEC, following Johnson et al. (2015) [32].

For phylogenetic typing, classic seven-gene multi-locus sequence typing (MLST) (v. 2.16.1) [33] was performed with assembled genomes using automatic scheme detection. Additionally, core genome MLSTs (cgMLST) were assigned by submitting raw reads to the Center for Genomic Epidemiology website (http://www.genomicepidemiology.org/, (accessed on 10 February 2022)) using cgMLSTFinder (v. 1.1), which runs KMA [34] against a chosen core genome MLST (cgMLST) database, here for *E. coli* [19]. The strains were assigned to the *E. coli* phylogroups using EzClermont (v. 0.6.3) (https://github.com/nickp60/EzClermont, (accessed on 10 February 2022)).

Genoserotyping was performed with the tool SeroTypeFinder (v. 2.0) [35,36]. For ambiguities in the O and H antigen classifications, final assignments were conducted according to the following criteria: (i) O9/O9a: O9, because the *wbdA* gene contained a cysteine at position 55 [37]; (ii) O9/O160: O160, based on the sequences of *wzx* and *wzy*; (iii) O13/O129: O13 [37]; O17/O77/O106 and O17/O73/O77: O77-group, because the serotype cannot be determined based solely on sequence data [37,38]; O18/O18ac: O18 [37]; O173:H4/H40: O173:H40, because the H4 allele is located in a 3 kbp long contig, while the H40 allele is in a contig spanning 30 kbp with an intact upstream region. O8/O32: O32, based on the sequences of *wzx* and *wzy*. An O8-classifying sequence, which is otherwise represented by *wzx, wzy, wzm* and *wzt* [35,39], was identified solely for *wzt*. The Platon plasmid analysis tool was used to discriminate plasmid and chromosomal origin contigs [40]. Whole-genome phylogenetic tree images were generated with ITOL (v.6) (https://itol.embl.de/, (accessed on 20 March 2022)). 

### 2.3. Closure of the Gap in the Plasmid p20E0407A

The primers flanking the gap (20E0407-hypPF: CGAAATCCGCAGCATGGC; 20E0407-klcA1R: GACAGGTTTCGCATATTGC) were synthesized by Eurofins Genomics. They were used in a standard endpoint PCR (25 cycles) with OneTaq DNA polymerase (New England Biolabs GmbH, Frankfurt/Main, Germany) following supplier recommendations and using 5 ng of purified genomic DNA of strain 20E0407 as template, an annealing temperature of 57 °C and an extension time of 30 s. PCR fragments were purified with the NucleoSpin Gel and PCR Cleanup Kit (Macherey-Nagel, Düren, Germany). An aliquot was mixed with one of the amplification primers and sequenced by Eurofins Genomics (Ebersberg, Germany). Sequence analysis was performed with Geneious Prime (v. 2021.0.1; Biomatters, Ltd., Auckland, New Zealand).

## 3. Results

### 3.1. Phylogenetic Analysis

The 39 *E. coli* isolates analyzed displayed high genomic diversity. They were distributed in five clusters corresponding to Clermont phylogroups [41,42] (Figure 1, thick colored line). Phylogroup B1, commonly associated with an animal origin [43], was identified most frequently. The sole isolate belonging to phylogroup C was also located in the B1 cluster. The second largest cluster was represented by phylogroup A, described to be more often found in humans [43]. Three remaining clusters consisted of a few strains belonging to phylogroups D, E and G. A total of 28 different MLSTs (Warwick scheme) were identified in the 39-strain collection, including 4 novel STs (13,045–13,048; Figure 1) [33]. The predominant MLSTs were 162 (5), 10 and 58 (3 each), followed by 533 and 1431 with two isolates each. 

The 39 strains also showed high diversity in their genoserotypes, featuring 35 different serotypes. Only four serotypes were represented by two strains (Ont:H15; O101:H9; O29:H10; O160:H19) (Figure 1).

To determine whether this high degree of genomic diversity reflected the overall diversity of the *E. coli* population [19], we compared our strains with the strains of the ECOR collection [44,45] (Figure 2). The strains were distributed throughout the phylogenetic tree, but as indicated previously, most of the strains were mixed with the ECOR strains belonging to phylogroups B1 and A. 

### 3.2. Virulence Factors

In total, fifty-one different virulence genes were detected (Appendix A). The genes most frequently identified were *terC* (tellurium ion resistance protein) in 97% of the strains (38/39), *ompT* (outer membrane protease) and *iss* (increased serum survival) in 74% (29/39) of the strains, followed by *traT* (outer membrane protein complement resistance) (69%; 27/39) and *gad* (glutamate decarboxylase) (66%; 26/39), indicating a low virulence potential for most isolates (Appendix A). 

According to the results of the search for the genes *chuA*, *fyuA*, *vat* or *yfcV*, which were used to classify uropathogenic *E. coli* strains (UPEC) [31], none of the strains scored as UPEC. Strain 20E0496 could potentially be UPEC because it carries five *afa* adhesin encoding genes, *iroN* (enterobactin siderophore receptor protein), *iucC* (aerobactin synthetase), *ompT*, *papA_F11* (major pilin subunit F11), *yfcV* (chaperone-usher fimbria gene), *traT* and *sitA* (iron transport protein) (Appendix A). There is no single genetic profile associated with an ExPEC/UTI (urinary tract infection) phenotype [36]. Five strains, 20E0407, 20E0410, 20E0446, 20E0579 and 20E0583, contained two or three of the genes *afa*/*dra*, *iutA*, *kps*MT II, *papA/papC* and *sfa*/*foc* (Appendix A) and were therefore designated putative ExPEC [30].

An atypical EPEC strain, 20E0539, was found (Appendix A). However, virulence factors typical of other *E. coli* pathovars, such as EAEC, EIEC, STEC or enterotoxigenic *E. coli* (ETEC) [36,46,47], were not detected in these isolates. Diffusely adherent *E. coli* (DAEC) and adherent invasive *E. coli* (AIEC) are reported to lack clear virulence factor signatures and are rather identified phenotypically [46], which was not performed in this study. 

### 3.3. Antimicrobial Resistance

All 39 *E. coli* isolates, obtained from fecal samples from 24 holdings, carried at least one antimicrobial resistance gene, and 59% (23/39) of the strains were multi resistant, carrying genes mediating resistance to three or more different antibiotic classes (Figure 3). In total, 28 different genes encoding for resistance to the different antibiotic classes of β-lactams, fluoroquinolones, folate pathway antagonists, aminoglycosides, tetracyclines, phenicols, macrolides and rifamycin were detected. Furthermore, three different mutations were detected in the *gyrA, parC* and *parE* genes. Antibiotic resistance was directed more frequently against β-lactams (35/39; 90%), followed by fluoroquinolones (62%, 24/39) and folate pathway antagonists (22/39, 56%,) (Appendix A). The corresponding resistance genes most frequently identified were β-lactamases, such as *bla*_TEM-1B_ (44%), ESBL enzyme encoding genes, such as *bla*_CTX-M-1_ (41%), and the aminoglycoside resistance-mediating gene *aph*(6)-Id (38%) (Figure 3). More than half of the isolates (54%) carried a mutation in *gyrA* and 49% an additional mutation in *parC*, reflecting, like the β-lactamases, the initial selection of isolates from enrofloxacin- or ceftiofur-containing agar plates. Resistance to tetracyclines by drug efflux was also observed (18/39; 46%), as was resistance to phenicols (12/39; 31%). The macrolide-resistance-mediating genes *mef*(C) and *mph*(G) were only identified in two strains and the rifampicin resistance gene *arr*-2 in a single strain. Genes that mediate resistance to colistin were not identified.

One strain, 20E0552, contained the rare *arr*-2 gene (Figure 3), an ADP-ribosyltransferase associated with resistance to rifampicin [48]. The contig encoding *arr*-2 contained four additional resistance genes, i.e., (i) *aad1*, mediating resistance against spectinomycin and streptomycin, (ii) *bla*_OXA-10_, mediating resistance against several β-lactams, including piperacillin/tazobactam, (iii) *cmlA1,* mediating resistance against chloramphenicol, and (iv) *dfrA14*, mediating resistance to trimethoprim (Figure 4). This gene group appears to be a resistance cassette, as it leads to 28 hits with 100% identity in the NCBI nucleotide database, all in *Enterobacteriaceae* (*Escherichia* spp. (n = 20), *Klebsiella* spp. (n = 6), *Salmonella* sp. (n = 1), *Yokenella regensburgei* (n = 1)) and all located on plasmids. In these plasmids, the cassette is embedded in different genetic contexts and flanked by genes associated with mobile genetic elements, such as integrases and transposases. 

Two strains, 20E0421 and 20E0503, isolated from different farms, contained the rare tandem gene combination *mef*(C)/*mph*(G) (Figure 3), which mediates high-level macrolide resistance through the action of a phosphotransferase and an efflux protein from the major facilitator superfamily [49]. It was recently identified in an STEC strain from France where it was located on a 202,201-bp plasmid with IncHI1A, IncHI1B(R27) and IncFIA(HI1) incompatibility groups. The additional resistance genes in that plasmid differ from the resistance genes identified in the two isolates from SAC [50]. In strain 20E0421, the *mef*(C)/*mph*(G) resistance element is located on contig44, which also contains an IncI1α (ST7, CC-7) identifier. The resistance element on contig42 of strain 20E0503 does not contain a plasmid replicon identifier.

The *bla*_CMY-2_ allele was identified in two strains (Figure 3). In strain 20E0407, the *bla*_CMY-2_ gene was present in contig15, together with an IncB/O/K/Z origin of replication. Its *inc*RNAI region identified the plasmid as belonging to the IncK2 subgroup, and the entire contig (88,592 nt) displayed strong similarity to plasmid pDV45 (85,963 nt; [51]). In strain 20E0402, the *bla*_CMY-2_ gene was located on contig 136 (9938 nt) featuring a common genetic environment with an upstream IS*Ecp1* element and downstream *blc* and *sugE* genes but without a plasmid replicon identifier.

The sequence of contig52 (20,749 nt; 20E0509), containing a *bla*_SHV-12_ gene frequently found in Europe [52], was identical (2 nt mismatch, 1 nt gap) to a region of the IncI1 plasmid pCAZ590 (117,387 nt; [53]), representing a multi-resistance gene cassette (*clmA1, aadA* (2x), *sul2*, *qac*). An IncI1 plasmid replication origin was detected in the genome sequence of strain 20E0509, but it was not present in the same contig.

### 3.4. Plasmids

At least one plasmid replicon was identified in 95% (37/39) of the isolates (Appendix A). One to seven different incompatibility groups (IncF, IncC, IncN, IncH, IncX, IncB and IncY) were detected in 72% (28/39) and Col plasmids in 61% (24/39) of the isolates (Appendix A). The Inc group most frequently identified was IncF (72%; 28/39), which consisted of different replicons (e.g., IncFIB and IncFIC), followed by Incl (43%; 17/39). The plasmid replicons IncF, Incl1, IncN and IncH1 were further subtyped with specific pMLST schemes [34]. Multi-replicon plasmid contigs containing IncFIB.AP001918.1 and IncFIC.FII_1 featuring the same ST (F89:B10) and alleles (FIB_1, FIC_4, FII_18) were detected in five isolates (Appendix A). 

An IncI1α plasmid replicon was identified in the same contig that encodes a *bla*_CTX-M-1_ allele in 12 out of 16 isolates. The contigs from five of these strains were classified as belonging to ST58, three as ST3, two as ST49, one as ST7 and one was not typeable (Appendix A). Five of the twelve strains had contig lengths that suggested they could represent full-length plasmids. Therefore, we checked these for sequence identities to continuous regions in other IncI1α plasmids and for overlaps at their 5′ and 3′ ends. This allowed us to perform ring closure, due to identical overlapping sequences, for one IncI1α/ST3 plasmid and the two IncI1α/ST49 plasmids. The latter two were sequence identical, except for a 12 nt insertion in a hypothetical protein of the plasmid from strain 20E0514 compared to the plasmid from strain 20E0498. One strain, 20E0519, featured the *bla*_CTX-M-1_ allele in the same contig as an IncN (ST1) replicon; the other three strains did not contain a resistance gene and a replicon identifier in the same contig. 

The contig encoding *bla*_CTX-M-15_ and a pO111-type *repA* gene was nearly identical in length and sequence to two plasmids deposited in the NCBI nucleotide database (accession numbers: MW646302 and CP075059), also allowing ring closure. The plasmid showed only four SNPs in toto with the two plasmids from the database, with the plasmid pERB8a1 (MW646302) featuring an additional 126 bp in frame deletion in a putative tail fiber protein-encoding gene. The ensuing 101,922 bp large plasmid carried only a *bla*_CTX-M-15_ resistance gene and no virulence factors.

The contig encoding *bla*_CMY-2_ and the IncK2 origin of replication was very similar in length and sequence to the plasmids pDV45 [51] and pCOV9 [54]. Sequence analysis using Geneious software suggested that a 29 nt gap was present in the contig. PCR primers flanking the gap were designed; a PCR fragment corresponding to the proposed length of 598 bp was obtained and sequenced to confirm the sequence and close the gap. The resulting plasmid p20E0407A was 88,620 bp long, contained only *bla*_CMY-2_ as a resistance gene, a *traT* gene as a virulence factor and a *psiBA* operon, known to inhibit the induction of the SOS response [55], with PsiA lacking the last 15 residues due to the insertion of a hypothetical protein ORF.

The 17 *bla*_TEM-1B_ encoding contigs were only rarely associated with a plasmid replicon identifier. One, from strain 20E0502, was classified as IncX1, and three others, from strains 20E0508, 20E0542, 21E0005, as IncF (Appendix A).

### 3.5. Co-Localization of AMR Genes and Virulence Factors on Plasmids

Most of the acquired AMR genes and virulence factors identified were classified as plasmid-encoded (Figure 5) by the Platon software, a tool that detects plasmid-associated contigs within bacterial draft genomes based on the distribution biases between the chromosome and the plasmid observed for specific protein-coding genes [40]. Notable exceptions among the AMR genes were *bla*_CTX-M-15_ with an ESBL phenotype, *floR* mediating phenicol resistance and *tet* genes encoding resistance to tetracyclines. At least half of the contigs containing these genes were assigned to a chromosomal location. Virulence factor-encoding genes are less likely than AMR genes to be plasmid-encoded, but of the 21 virulence factors identified, only 6 (*cba*, *cea*, *iss*, *ompT*, *papC*, *tsh*) were assigned to a chromosomal location equally or more frequently than to a plasmid location.

### 3.6. Comparison of ST162 Isolates with Enterobase Isolates

The most frequent sequence type found in the SAC strain collection was ST162 (five isolates). Sequences of these five isolates were compared to 721 ST162 genomes present in Enterobase (accessed on 23 February 2022; [19]) and their total and core SNP differences determined. Central German SAC isolates are located in different branches in a small section of the tree (Figure 6a). Each respective closest neighbor shares the SAC isolate’s genoserotype. The metadata on the source, year (2017 to 2019) and country of isolation (Appendix A) revealed that three strains were associated with a human clinical source and two strains with a poultry host. Human isolates were sampled in the UK and France, while the poultry isolates were from Hungary. One clinical strain was classified as UPEC (JB7842AA). Database sequences differ by 49–205 total SNPs or 10–43 core SNPs from the respective nearest SAC strain. Regarding the virulence factors and AMR genes (Figure 6b), the two poultry strains and one human strain from the UK have identical gene presence/absence profiles. The French UPEC strain lacks two biocin-associated genes (*cvaC*, *mchF*) but contains additional AMR genes directed against β-lactams, aminoglycosides, sulfonamides and tetracycline, while the second clinical strain from the UK (AC5385AA) features additional virulence factors encoding iron acquisition, biocins and increased serum survival, as well as resistance genes against aminoglycosides, β-lactams, trimethoprim/sulfonamide and tetracycline compared to the strain of the SAC. 

### 3.7. Comparison with Other Alpaca and Llama Isolates

Enterobase contained 17 whole-genome sequences of *E. coli* strains isolated from llamas and alpacas from Peru and the US that were not STEC. We compared these using total SNP differences with the central German SAC sequences to determine if there were specific geographical clusters of strains (Figure 7).

SAC strains from the American and German SAC groups cluster together with their respective phylogroups. Strains with identical sequence types are located together, and American and German strains are mixed throughout the entire phylogeny tree and do not display geographic-location-dependent clustering. 

## 4. Discussion

SAC are becoming increasingly popular in Germany [3,4] but are not subject to recording of their health or microbiological status. They may carry different zoonotic agents and potentially transmit them to other livestock species and humans. A robust and comprehensive knowledge base on epizootic and zoonotic AMR pathogens in SAC is urgently required to draft appropriate future surveillance studies and to aid in the development of preventive medicine measures for improving both animal and human health. A recent pilot study among SAC from central Germany investigated the prevalence of (i) a set of bacterial pathogens and (ii) resistance to fluoroquinolones and cephalosporins (third/fourth generation) in indicator *Escherichia coli* [4]. The study presented herein focused on the population structure of those antimicrobial-resistant *E. coli* isolates. As a commensal member of the vertebrate gut microbiota, *E. coli* is the most common aerobic bacterial species [43] and also serves as an indicator organism for AMR in several surveillance programs [56]. Due to its remarkable genomic plasticity [57], it can easily acquire and lose mobile genetic elements encoding virulence factors and AMR genes, subsequently leading to the emergence of many different pathotypes [15]. To gain a first impression of the population structure of AMR *E. coli* in SAC, a selected set of isolates was characterized using short-read WGS for the detection of AMR genes, clonal lineages and pathovars associated with high-risk clones at the international level.

### 4.1. The E. coli Strains Isolated from the Central German SAC Are Phylogenetically Diverse

Phylogenetic analysis revealed a high diversity of strains with respect to phylogroup, sequence type and genoserotype (Figure 1), which became particularly apparent when compared to the ECOR collection [44,45], which also showed a broad distribution, with many strains belonging to phylogroups B1 and A. In these two phylogroups, representatives of both collections mix without forming distinct clusters. Additionally, we compared strains isolated from German SAC with sequences deposited in Enterobase from 17 non-STEC strains isolated from SAC in America (Peru and the USA) to verify clustering dependent on the geographic origin of the strains. Although the number of whole-genome sequences available from the American SAC was sparse, they did not cluster separately but were immerged among the SAC strains throughout the entire phylogeny tree. The large diversity observed here could be due to the small sample size, which does not yet allow the identification of potential physiologic and cellular preferences of *E. coli* strains in colonizing SAC. This is supported by the observation that the 17 isolates from American SAC contained only one strain with a sequence type (ST10) that was also found in the strains from the German SAC. The isolation and analysis of additional strains, including those from other geographic regions, will be necessary to obtain a more complete picture of the global distribution of *E. coli* in these animals. 

In strains isolated from the German SAC, the most frequently found sequence types were 162 (n = 5), 10 and 58 (each n = 3). These are all globally disseminated sequence types. ST162 strains can cause disease in birds, companion animals and humans/children [58]. Because they are associated with several different β-lactamases [59] and described to belong to a globally disseminated clone [59], representatives of which can be isolated from many different sources, we compared our five isolates with 721 ST162 genomes present in Enterobase. The five SAC isolates differed in a relatively small number of SNPs compared to their closest neighbor strains, which were isolated from human clinical sources or poultry. However, the differences in virulence factors and AMR genes did not correlate with the differences in SNP counts. This shows that the ST162 strains of SAC can spread and may have been obtained from different vertebrate hosts, losing and acquiring virulence factors and AMR genes during this process. There appears to be a trend toward a higher number of AMR genes in the isolates from human clinical samples, but the analysis of additional human/animal pairs will be necessary before any clear patterns might emerge.

ST10, assigned to three strains (Figure 1), is a globally distributed strain commonly isolated from a wide diversity of hosts, environments and regions [60]. It is one of the most frequently represented sequence types in Enterobase and is often an intestinal commensal inhabitant that lacks virulence-associated genes required for pathogenesis [19,61]. One strain, 20E0508, was highly resistant, featuring ten AMR genes. The other two strains carried only one or three resistance genes, respectively. A similar divergence was also observed in strains isolated from veal calves. VirulenceFinder software identified one (20E0424: *terC*), three (20E0508: *hra*, *papA*_F19, *terC*) and five (20E0498: *astA*, *capU* (hexosyltransferase homolog), *ompT*, *terC*, *traT*) virulence genes, overall indicating low pathogenic potential, despite the fact that *astA* encodes the enteroaggregative heat-stable enterotoxin EAST1, *hra* a heat-resistant agglutinin and *papA* the main component of P-type fimbriae [29,35].

Strains classified as ST58 are among the top 20 ExPEC strains. They are regularly isolated with or without selection for antibiotic resistance [62] from healthy and diseased humans, livestock, companion and wild animals [63]. The ST58 strains sequenced herein were not classified as ExPEC, but ExPEC strains are difficult to discriminate solely by molecular markers [61]. A recent genomic survey of 752 ST58 strain sequences identified six clusters with two major ones, one containing mainly strains from bovine sources and the other frequently containing ColV F plasmids [63]. According to the Liu criteria [64] applied in that genome analysis, only strain 20E0421 seems to contain a ColV plasmid. The mapping of its assembled contigs to the archetypal ColV plasmid pCERC4 revealed approximately 60% coverage of the entire plasmid, with a gap spanning the ColIa element and a large part of the transfer region. The strain also encodes *fyuA*, a marker for the pathogenicity island of yersiniabactin siderophore [65]. However, in contrast to the ColV+/BAP2 strains described in Reid et al. [63], this isolate did not contain more ARGs and virulence factors than the other two ST58 strains. Strain 20E0490 was multi-resistant with 13 AMR genes mediating resistance to seven antibiotic classes, but, harboring only four virulence factors, most likely classifies as commensal.

### 4.2. The Load of Virulence Factors Indicates a Low Pathogenic Potential

In strains isolated from SAC, more than 50 different virulence factor genes were detected, but no high-risk pathovars, such as ETEC or EAEC, were identified. STEC strains were isolated from SAC [4,16] (see also Enterobase, e.g., BioProject PRJNA579481) but were not included in this survey. One strain, 20E0539, was classified as atypical EPEC; another strain, 20E0496, could potentially be designated as UPEC. Five strains were putatively assigned as ExPEC [30] but were not associated with a urinary tract infection phenotype. None of them belonged to the top 20 ExPEC STs [62]. The counts of individual strains ranged from encoding 1 virulence factor to a maximum of 25, with a mean value of 10.3 ± 6.2 and a median of 10 (8.1–11.9 95% CI). Even the apathogenic K12 laboratory reference strain MG1655 harbors four virulence genes. The pathogenic potential of the *E. coli* strains isolated from German SAC and phenotypically resistant to at least one of the antibiotic classes deployed during isolation seems to be quite low, lower than in other farm animals [66]. Many of the virulence factors identified appear to be related to cell adhesion (15/51), biocin production (11/51) and siderophores (7/51). Nevertheless, one should keep in mind that ST162 strain 20E0410 contains the same virulence factors plus two biocin-associated genes as strain JB7842AA, which was isolated from a French patient with a urinary tract infection. Two other ST162 strains have identical virulence factor profiles to two clinical isolates from the UK. Therefore, a final assessment of the virulence potential of SAC strains is difficult, as it will in part depend on the vulnerability of the infected host and other predisposing factors. 

### 4.3. The AMR Genes Are Unevenly Distributed between Strains

Little information is available on the phenotypic antimicrobial resistance in *E. coli* from SAC [12,13,67,68]. However, no WGS analysis of AMR genes in SAC has been published for *E. coli;* only two studies address resistance genes in *Enterococcus* spp. [69] and in MRSA [70]. Therefore, the analysis of AMR gene presence in the *E. coli* population of SAC in this study was used as a surrogate to estimate the presence and diversity of AMR genes in Gram-negative bacteria. Because the initial strain isolation protocol [4] involved selection for ceftiofur or enrofloxacin resistance, this could have introduced a bias toward higher total AMR gene counts than if a non-selective strain isolation protocol had been used.

Our results show a high diversity of AMR genes—twenty-eight different resistance genes and three different chromosomal mutations (*gyrA*, *parC* and *parE*) in total. The distribution of AMR genes showed a biphasic profile, in which 40% of the strains carried fewer than three genes mediating resistance against different antibiotic classes, and the remaining 59% (23/39) of the strains were multi resistant [71], with 23% of the strains carrying between 10 and 14 resistance genes. 

The most common resistance genes were β-lactamases (in 35/39 isolates), such as *bla*_TEM-1B_ (44%), which is one of the most widely distributed β-lactamases in the world [72], and *bla*_CTX-M-1_ (41%), commonly isolated from animals in Europe [73]. Furthermore, *bla*_CMY-2_ was identified in two strains. This gene is the most frequently detected *ampC* gene encoded by plasmids in *Enterobacteriaceae.* It is usually found in Europe in strains isolated from poultry [73]. Interestingly, only five strains contained both a *bla*_TEM-1B_ gene and another β-lactamase gene. Whether this is only due to the resistance genes being located on different plasmids or whether any counterselection against the dual β-lactamase gene presence is active is not clear so far and will require further analysis. 

Fluoroquinolone antibiotics have been widely used in livestock, making food animals an important reservoir of resistance [74,75]. In swine, a high prevalence of resistance was strongly correlated with the use of fluoroquinolones [76]. The resistance-mediating mutations found are common in *E. coli* in STs and clonal complexes distributed worldwide, such as CC10 (ST10, ST48 and ST744) [76] or ST410 and ST162, which are both present in the SAC strain collection and carry these chromosomal mutations. In addition to the high prevalence of these resistance mutations originating from the initial selection process [4], a high prevalence of resistance to tetracyclines (46%), trimethoprim/sulfonamides (31% *dfrA* + *sul*, 53% *dfrA* or *sul*) and phenicols (31%) was detected. The European Medicines Agency’s “Categorisation of antibiotics used in animals promotes responsible use to protect public and animal health” recommends in category D (prudence) the use of amoxicillin/ampicillin, tetracyclines and trimethoprim/sulfonamides as first-line treatments (https://www.ema.europa.eu/en/documents/report/categorisation-antibiotics-european-union-answer-request-european-commission-updating-scientific_en.pdf (accessed on 15 March 2022)). Resistance to amphenicols, which are classified as category C (caution; to be considered only when there are no clinically effective antibiotics in category D), is mediated in six strains by *floR*, which confers resistance to florfenicol. This antibiotic is used routinely in veterinary medicine, in contrast to chloramphenicol, for which ARGs (*cat* and *cmlA*) were detected in seven strains. The prevalence of resistance to macrolides (category C) was low, and resistance to colistin (category B; restrict was not detected, indicating possible treatment options.

Interestingly, several resistance genes rarely found in *E. coli* were also detected in strains isolated from SAC. The *arr*-2 gene was detected in one strain (20E0552) as part of a highly conserved and mobile resistance cassette located in plasmids found in *Enterobacteriaceae*, and the tandem gene combination *mef(C)/mph(G)* was detected in two strains (20E0421 and 20E0503). We searched for homologous Enterobacterales plasmids and for *mef(C)/mph(G)* genes with 100% coverage and homology in all *E. coli*/*Shigella* sequences in the NCBI databases “Nucleotide collection (nr/nt)” and “RefSeq Genome database”. The sequence of contig44 of strain 20E0421 was close to numerous IncI1 plasmids described in Enterobacterales, but none of these plasmids carried the *mef(C)/mph(G)* genes. Additionally, we identified these genes in two strains (Win2012_WWKa_NEU_19 and KPC1628) deposited in GenBank as unpublished sequences isolated from German wastewater or from a human clinical sample from Brazil. However, in these samples, the respective contig containing *mef(C)/mph(G)* did not contain a plasmid replicon identifier. It is intriguing that a third sample was found in central Germany, albeit in wastewater. A recent publication [50] identified the tandem gene pair in an STEC strain (isolate 45466) and additionally in six *E. coli/Shigella* isolate sequences from public databases, one of which was also identified by us (Win2012_WWKa_NEU_19). They originated from all over Europe, mostly from aquatic environmental sources. There is no official information on antibiotic usage in SAC from Germany, but an inspection of the questionnaires from the farms did not reveal treatments with macrolides. The possibility that the SAC might have ingested resistant bacteria from local water sources during trekking tours cannot be excluded.

### 4.4. The AMR Genotypes Correlate Well with the Phenotypes of the Strains

Comparing the WGS results with phenotypic data previously published [4], we detected a perfect correlation between the resistance phenotype and the presence of one or more genes mediating resistance to tetracyclines and gentamicin. Resistance against Co-trimoxazol was observed when both genes (*dfr* and *sul*) were present, and isolates were susceptible when none of them was found. However, several strains were resistant when only one of the genes was detected. The presence of an *aadA* gene was found to coincide with a resistant or intermediate phenotype against spectinomycin in 13 out of 14 strains. One strain (20E0607) was resistant to spectinomycin without carrying an *aadA* gene; one strain carrying an *aadA* gene (20E0506) was susceptible to spectinomycin. Resistance to ampicillin was detected in all strains encoding one or more β-lactamase genes. All strains encoding a *bla*_CTX-M_ (-1, -15, -27, -65) or a *bla*_SHV-12_ allele (21/22) were resistant to ceftiofur and cephalothin, with one exception. The strain 20E0402 was susceptible to ceftiofur despite the presence of a *bla*_CTX-M-1_ allele. Only a few strains without a resistance gene or with TEM β-lactamase showed an intermediate phenotype toward cephalothin, and no strains were resistant to amoxicillin/clavulanic acid. The presence of one or more mutations in both *gyr/par* genes resulted in resistance to quinolones, as described [75]. Only one isolate (21E0005), which carried both mutations, displayed a susceptible phenotype. Taken together, the presence of a resistance gene correlates well with the resistance phenotype in the SAC strain collection analyzed herein.

### 4.5. Conserved Plasmids Are Identified despite Apparent Plasmid Heterogeneity

Plasmids are the predominant mobile genetic element responsible for intercellular transmission of genes encoding antimicrobial resistance, virulence factors and other niche-adaptive traits, thus contributing to bacterial ecology and evolution [77,78,79]. This is evidenced by the results of the Platon analysis, unveiling a plasmid location for most of the AMR genes and virulence factors identified. 

Plasmids can spread horizontally within or between a bacterial species by conjugation or mobilization. Except for Col plasmids, which were not analyzed in detail here, the replicon types identified by PlasmidFinder represent low-copy plasmids present either in *Enterobacteriaceae* (IncB, IncF, IncH, IncI, IncX, IncY) or commonly found in *E. coli* but with a broader host range (IncC, IncN). Since the original screening step [4] was for resistance to ceftiofur, a third-generation cephalosporin, we assessed contigs containing β-lactamase genes for replicon identifiers. The ESBL gene most frequently found was *bla*_CTX-M-1_, which is widely distributed among livestock in Europe [73]. In 12 out of 16 strains, it is located on an IncI1α plasmid—also a frequent combination in European livestock [80,81]. These IncI1α plasmids are not all identical; the pMLST analysis identified five different sequence types (Appendix A). Only 4/12 isolates carried plasmids with the major sequence types pST3 and pST7 [80]; the remaining 8 strains carried plasmids from three minor STs (Appendix A). Five of these were classified as pST58 and were isolated from strains with four different types of sequence types and originated from four different farms, indicating a broad dissemination in SAC from Central Germany. 

We analyzed the plasmids for which we achieved ring closure in more detail. The IncI1α/pST3 plasmid p20E0605A displays similarity in a BLAST analysis to ten plasmids with 100% coverage and identity (nr/nt database; accessed 27 June 2022). It is very similar to plasmids identified in *E. coli* strains isolated from a French river [82], such as pESBL26. Both carry the same combination of resistance genes of *bla*_CTX-M-1_, *sul2* and *tet*(A). Their aligned sequences differ by one distinct translocation of 551 bps carrying a hypothetical ORF, four SNPs, a single nucleotide insertion and four deletions of twice one nt, 117 or 171 nts length over approximately 107,000 nts. Similarly, the IncI1α/pST49 plasmids p20E0498A and p20E0514A are identical to the plasmid pECOH8 (unpublished; accession no. HG739083), except for the 12 nt deletion in p20E0498A. Likewise, the plasmid p20E0407A with a *bla*_CTX-M-15_ gene and a pO111-type *repA* gene differs by only four SNPs from the plasmid pCTX_B2_4, isolated in China (unpublished, accession no. CP075059). The IncK2 plasmid containing the *bla*_CMY-2_ allele is very similar to the plasmid pDV45, isolated in Switzerland from poultry meat [55], containing only eight SNPs, three small indels of 20, 8 and 1 nt length and a larger insertion of 2646 nts encoding IS*21* transposase sequences.

Less information is available for the other β-lactamase-gene-encoding contigs. A more detailed analysis of their sequence surroundings will require long-read sequencing to resolve repeats in the respective sequences. We would like to point out that IncN/pST1 plasmids that encode a *bla*_CTX-M-1_ gene have been identified in isolates acquired from aquatic environments [82] or wild animals [83].

Taken together, antimicrobial-resistant *E. coli* strains isolated from SAC display a variability typical for plasmids commonly found in *Enterobactericeae* [52], with several highly conserved members.

## 5. Conclusions

This first pilot study of the population structure of antimicrobial-resistant *E. coli* of SAC in Central Germany revealed that SAC harbor many different phylogenetically diverse strains of *E. coli.* These can encode virulence factor profiles associated with human disease (atypical EPEC, UPEC and ExPEC), although such strains do not appear to be frequent in the animal population under study. The ARG profiles are biphasic, with approximately 40% of the isolates showing genes that mediate resistance to fewer than three classes of antibiotics, whereas about one-quarter of the strains possess ten and more resistance genes. The results from the ARG and plasmid analyses remarkably demonstrate both the dynamics and the persistence present in the plasmid and resistance cassette turnover. Taken together, the pathogenic risk and the AMR situation regarding *E. coli* from Central German SAC appear to be like that in other livestock, companion or wild animals. Therefore, similar sets of rules and regulations as in other livestock animals should be adopted in SAC for AMR monitoring and management.

## Figures and Tables

**Figure 1 microorganisms-10-01697-f001:**
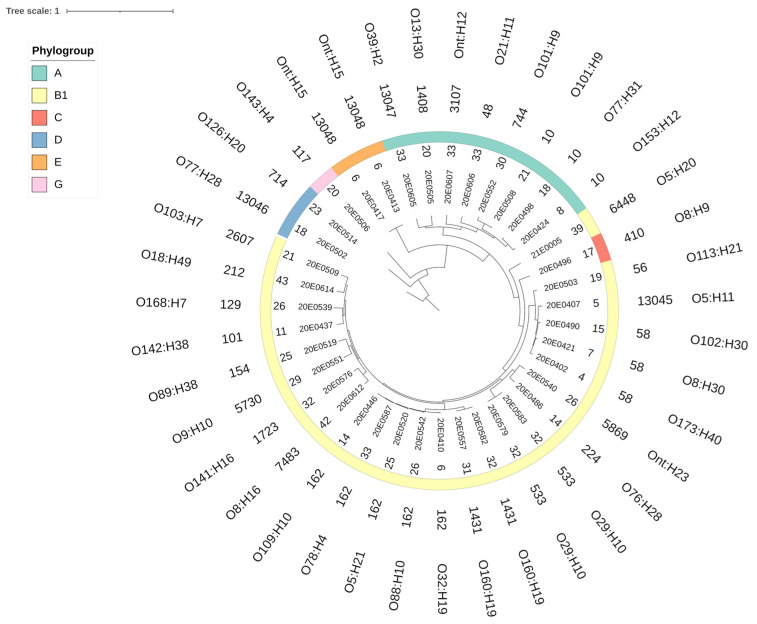
Whole-genome phylogenetic tree of the 39 *E. coli* isolates. Phylogenetic relationship of *E. coli* isolates based on core SNPs from whole-genome sequencing. Isolate and farm IDs are presented in the inner ring. The phylogroups are represented by different colors in the middle ring. The next ring indicates the multi-locus sequence type (MLST; Warwick scheme); the outermost ring indicates the O and H antigen groups assigned by genoserotyping. The image was generated with ITOL (v. 6).

**Figure 2 microorganisms-10-01697-f002:**
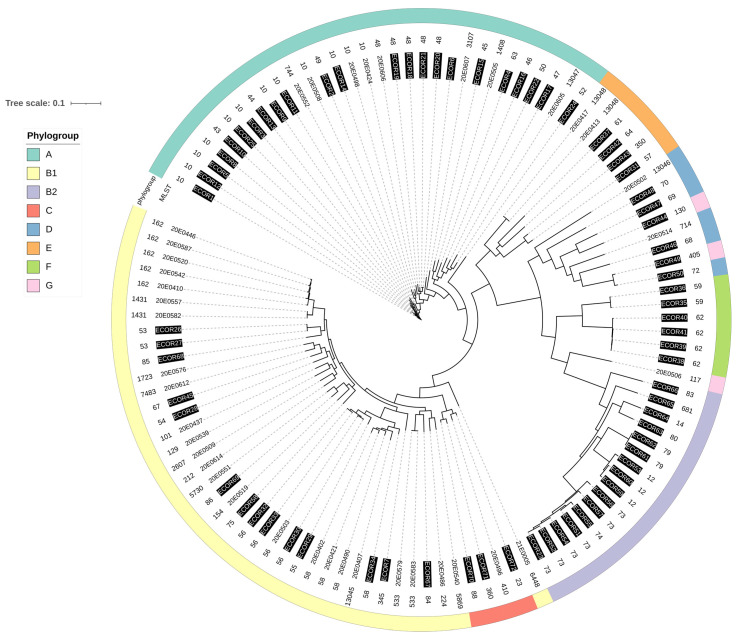
Whole-genome sequencing tree with the 39 *E. coli* strains from SAC and 72 ECOR strains (inverse print) in the innermost ring. The middle ring indicates multi-locus sequence typing (MLST) assignments; the outermost ring indicates the phylogroups in different colors. The image was generated with ITOL (v.6)).

**Figure 3 microorganisms-10-01697-f003:**
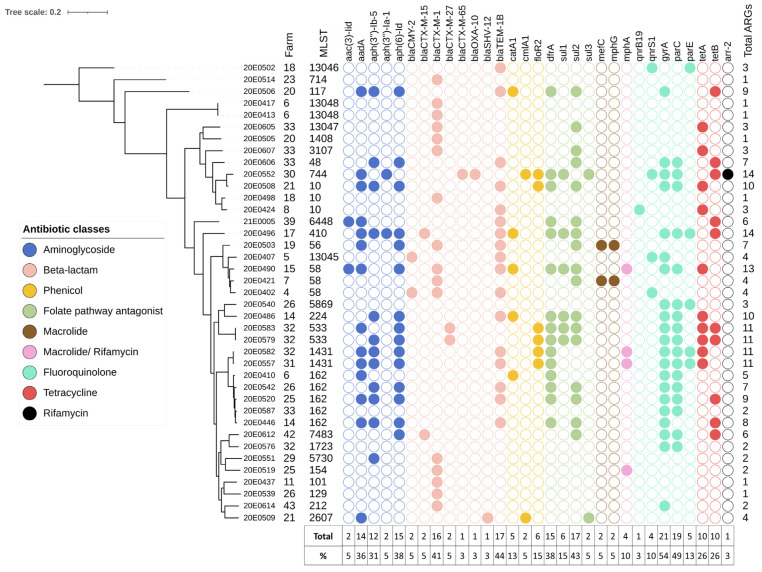
Whole-genome phylogenetic tree showing the distribution of the resistome of the 39 *E. coli* strains isolated from SAC and whole-genome sequenced. The strain designation, farm ID and MLST of the samples are indicated in the first three columns. Resistance genes, grouped by antibiotic class, are demarcated by colored circles. The last column indicates the total number of resistance genes per sample. The table below the plot shows the representation of the resistance gene, which contains the total number of each resistance gene detected in the sample and its percentage in all strains. The image was generated with ITOL (v.6).

**Figure 4 microorganisms-10-01697-f004:**
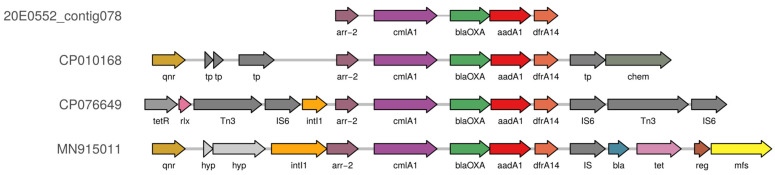
*arr*-2 resistance gene cassette from strain 20E0552 in its sequence context from different plasmids. The resistance gene cassette containing the *arr*-2 gene from contig78 of the WGS assembly for *E. coli* strain 20E0552 is shown. The identical resistance cassettes of three different *E. coli* plasmids taken from GenBank are shown beneath contig78 with their respective flanking sequences. The accession numbers of the plasmids are listed on the left. Genes are depicted as arrows, and their annotations are from the GenBank files. The image was generated with gggenes (v. 0.4.1).

**Figure 5 microorganisms-10-01697-f005:**
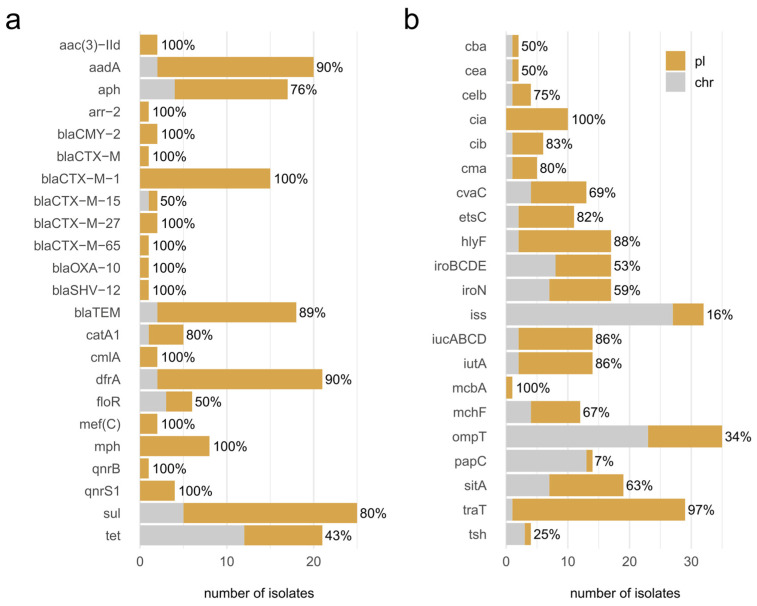
Platon analysis of the plasmid (pl.) or chromosomal (chr.) location of acquired (**a**) AMR genes or (**b**) virulence factors. The contigs encoding genes identified as mediating AMR or as virulence factors were analyzed for plasmid- or chromosome-typical markers in all strains isolated. The percentage value represents the fraction of plasmid-identified contigs for the respective gene.

**Figure 6 microorganisms-10-01697-f006:**
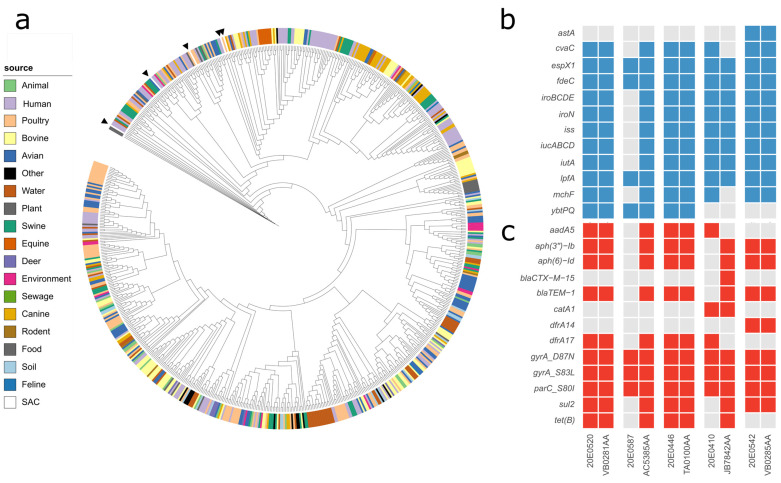
(**a**) Whole-genome sequencing tree of 721 *E. coli* ST162 sequences from Enterobase and five strains from the SAC collection, which are highlighted by black arrowheads. (**b**,**c**) Heat maps of the presence/absence of virulence factors (**b**) and AMR genes (**c**) ordered in pairs of the respective SAC strain and the nearest ST162 strain from Enterobase. The image was generated with ITOL (v.6).

**Figure 7 microorganisms-10-01697-f007:**
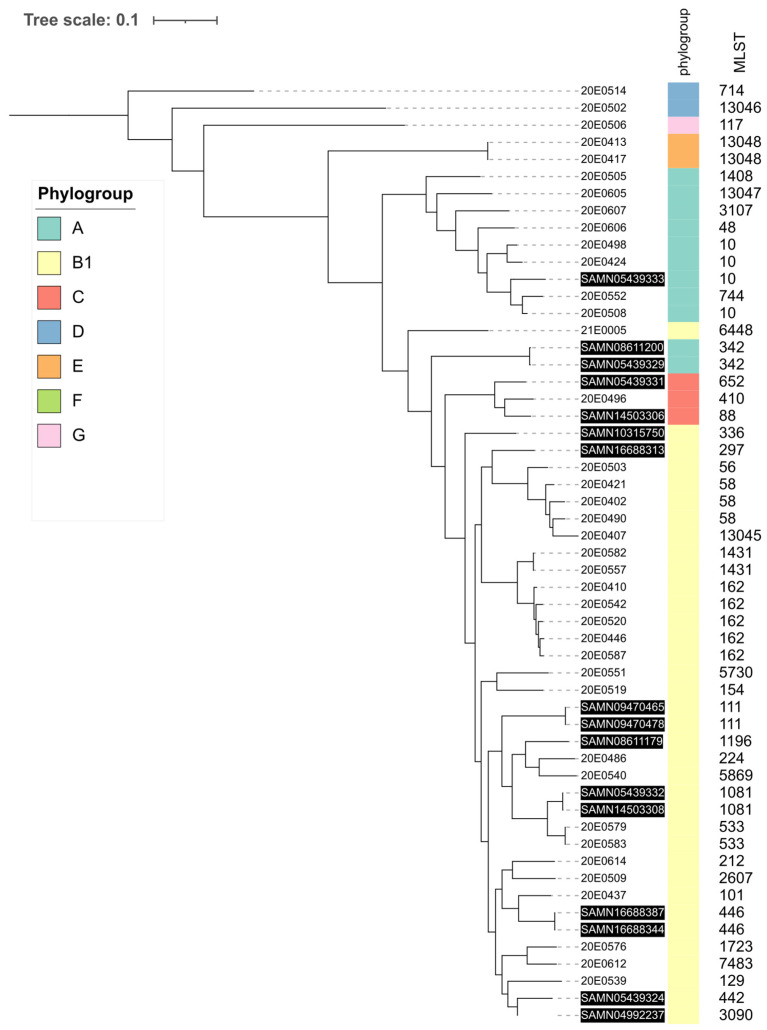
Whole-genome sequencing tree with the 39 *E. coli* strains of this study and 17 *E. coli* strains isolated from SAC in Peru and the USA and deposited in Enterobase (inverse type). The phylogroups are represented by different colors, and the multi-locus sequence typing (MLST) assignments are shown on the right. The image was generated with ITOL (v.6).

## Data Availability

The raw sequence data generated during the current study are available at https://www.ncbi.nlm.nih.gov/bioproject/PRJNA806394 accessed on 14 July 2022, The sample and accession numbers are listed in Appendix A.

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
