# Peer review of "Comparative Genomic Analysis of Antimicrobial-Resistant Escherichia coli from South American Camelids in Central Germany"

_microorganisms, 2022, doi:10.3390/microorganisms10091697_

Round 1
Reviewer 1 Report
The monitoring of antimicrobial resistance and virulence potential of Escherichia coli especially among strains derived from livestock species is very important and interesting from epidemiological point of view. The Authors investigated 39 E. coli strains from South American camelids living in holdings in central Germany. This study mainly based on whole genome sequencing of E. coli strains and bioinformatic analysis of the data.
Major comments
This is a well written manuscript. The study involved the characterization of the strain pool, which is not very large (39), but the analyzes are broad, very detailed, and provide valuable information on phylogeny, antimicrobial resistance, virulence factors, and the chromosome or plasmid location of the analyzed genes. In addition, the data was compared with data from other animal and human strains deposited in sequence databases. The results are presented in an interesting and good quality way. This is a very good job.
Specific comments
Introduction
Line 58
„……produced prevalence data for phenotypic resistance to cephalosporins in SAC similar to values ……” – should rather be: ……produced prevalence data for phenotypic resistance to cephalosporins among AMR E. coli from SAC similar to values…………
Materials and methods
Line 87
„The isolates had been extracted from composite fecal samples….. ” – extracted is rather incorrect term, should be, for example: E. coli strains had been isolated
2.1 Strain Selection
Line 85
„For this study, we selected 39 E. coli strains from a collection of 63 isolates…” - It is not clear what the selection criteria were? Why did the Authors select only 39 strains?
Line 91
Typing error
Should be: MLVA
Results
Virulence factors
Line 207
Typing error
Should be: kpsMT II
Author Response
Please see the replay in the attachment

Reviewer 2 Report
General comments
The study presented by the authors in this manuscript is of great values in identifying potential sources and reservoirs of AMR. However, there are issues that need to be addressed.
In the isolate selection, the authors mention that the isolates were selected based on their three criteria including e (i) drug resistance profile, (ii) MVLA type, and (iii) animal group from which they had been isolated. It would be good to expand on these criteria a little more. For example, considering the drug resistance profile, was the selection based on multidrug resistance? Was it based on resistance to the different antibiotic classes tested?. Similarly, how were the animal groups determined? Were the groups sampled on different dates? How were the groups determined?
Regarding sequencing, why were the isolates sequenced on different platforms? What guided the choice of what isolates were sequenced on what platform? Was this just a random assigning procedure was this just a logistic issue?
In the discussion, the authors have alluded the presence of some rare ARGs to the possible ingestion of resistant bacteria in water during the animals' tours (Lines 559-560). However, it would be good to have an idea of the use of antibiotics in these animals in Germany, as this could point to possible sources of the resistance observed.
Specific comments
Lines 146-147: This is hanging. Please read again and complete if applicable.
Author Response
Please see the replay in the attachment
